# A Preterm Infant with Feeding Aspiration Diagnosed with BOR Syndrome, Confirmed Case by Whole-Genome Sequencing and Structural Variant Calling

**DOI:** 10.3390/children10010076

**Published:** 2022-12-30

**Authors:** Da Hyeon Kim, Misun Yang, Heui Seung Jo, JongHo Park, JaHyun Jang, Sunghwan Shin, SeHyung Son

**Affiliations:** 1Department of Pediatrics, Samsung Medical Center, Seoul 06351, Republic of Korea; 2Department of Pediatrics, Kangwon University Hospital, Chuncheon-si 24289, Republic of Korea; 3Clinical Genomics Center, Samsung Medical Center, Seoul 06351, Republic of Korea; 4Department of Laboratory Medicine and Genetics, Samsung Medical Center, Seoul 06351, Republic of Korea; 5Department of Pediatrics, CHA Ilsan Medical Center, Goyang-si 10414, Republic of Korea

**Keywords:** branchio-oto-renal syndrome, preterm infant, whole-genome sequencing

## Abstract

Branchiootorenal (BOR) syndrome is a rare autosomal dominant inherited disease with a prevalence of approximately 1 in 40,000 newborns. This disease is characterized by hearing loss, preauricular pits, branchial fistulas or cysts, and renal dysplasia. We discovered a case of BOR syndrome in a premature 2-week-old female infant with a gestational age of 32 weeks and two days. She and her family had major symptoms and a family history of BOR. BOR syndrome was confirmed by whole-genome sequencing and structural variant calling, which revealed an *EYA1* exon 5–6 deletion. The infant had recurrent sleep and feeding cyanosis with second branchial anomalies. Via videofluoroscopic swallowing study and a modified barium swallow test, penetration into the vocal cords was observed before and during swallowing when bottle feeding. This is the first report of a preterm infant early diagnosed with BOR syndrome in which deletion margin was accurately identified by whole-genome sequencing and structural variant calling in Republic of Korea.

## 1. Introduction

In 1864, Heusinger used the term ‘branchial’ to describe a birth defect of the branchial apparatus of the neck, and comorbidities of this deformity, including hearing impairment, were reported [1]. In 1975 and 1978, Melnick et al. and Fraser et al. reported a disease accompanied by hearing loss, branchial anomalies, ear deformities, and kidney malformations. The disease was called branchiootorenal (BOR; MIM #113650) syndrome [2,3]. BOR syndrome is a very rare disease with an autosomal dominant inheritance and an estimated prevalence of 1 in 40,000 newborns [4]. Approximately 2% of children with severe hearing loss have BOR syndrome [4]. This disease is characterized by branchial fistulas with abnormal passages from the pharyngolaryngeal space to the surface of the neck, branchial cysts, hearing loss, and kidney anomalies.

## 2. Case Presentation

The patient’s gestational age was 32 weeks and two days, and the birth weight was 1710 g. An emergency cesarean section was performed due to preterm labor and preterm premature rupture of the membrane. Initial crying at birth was short and weak. At 1 min after birth, the heart rate was less than 100 beats with auscultation. Positive pressure ventilation was started with the NeoPuff. The heart rate recovered. Flow by O_2_ was given after positive pressure ventilation was terminated. Desaturation occurred, and neopuff PEEP was repeated. The infant was admitted to the neonatal intensive care unit (NICU) while applying PEEP. Her Apgar score was 5 points at 1 min (all points 1) and 8 points at 5 min (tone 1 point, reflex 1 point). Nasal continuous positive airway pressure was administered after admittance into the NICU (from day 1 to the morning of day 3 after birth). From the morning of day 3 after birth, High flow nasal cannula (HFNC) O_2_ was applied and O_2_ was stopped on the morning of day 5 after birth. For feeding desaturation and cyanosis, HFNC O_2_ was applied from day 18 to day 21 after birth and then discontinued. Feeding desaturation occurred during or after bottle feeding. From day 37 onwards, O_2_ nasal prong was applied only for feeding due to feeding desaturation.

The chief complaint was a pustule on the skin of the right neck, which was found incidentally at 2 weeks old. After removing the pustule with four cotton swabs, a pit was uncovered. A pit was also faintly seen on the left side. On the next day, cervical ultrasonography of the midportions of both sides of the neck revealed a tubular cystic structure on the medial side of the sternocleidomastoid muscle, extending to the cranial side. Therefore, second type branchial cleft fistulas on both sides were diagnosed. On day 18 after birth, a small amount of transparent viscous liquid came from the right neck pit. No redness, swelling, tenderness, or heat was observed around the pit. Extended spectrum beta-lactamase-positive Escherichia coli and methicillin-resistant Staphylococcus aureus were cultured from pus on the cotton swab at 2 weeks old. No bacteria grew in the blood culture test on day 18 after birth. Tazocin (piperacillin + tazobactam) was administered from day 18, as an empirical antibiotic. The initial and follow-up C-reactive protein tests were negative. No bacteria grew in follow-up pus cultures, so empirical antibiotics were stopped on day 23 after birth.

The patient exhibited bilateral branchial cleft fistulas and bilateral preauricular pits, and the father showed signs of surgery for branchial cleft cysts on both sides of the neck. The father stated that branchial cysts and fistulas were surgically removed from both sides of his neck in elementary school due to infection and that he had bilateral preauricular pits. His ears appeared different, and he stated that his hearing became a little weak in the high frequency range as he became an adult. He previously had facial nerve palsy and no kidney problems. The paternal grandmother had preauricular pits and poor hearing, which required a hearing aid. There was a history of surgery for bilateral branchial cleft cysts but no kidney problems. The patient’s great-grandmother also had preauricular pits, but her aunt was asymptomatic. According to the diagnostic criteria for BOR syndrome presented by Chang et al., if there is a family history, the diagnosis can be made when one or more of the main diagnostic criteria, including branchial cleft anomaly, deafness, preauricular pits, and renal anomaly, are meant [5]. Therefore, this case was suitably diagnosed as BOR syndrome; the patient had a family history of two types of branchial cleft anomalies and preauricular pits. Figure 1 shows the patient’s pedigree. Figure 2 and Figure 3 show the patient’s bilateral branchial cysts, fistula openings, and preauricular pits.

Oral feeding was attempted because the patient wanted to eat by mouth on day 2 and from around day 13 of life (34 weeks of corrected age). Oxygen saturation decreased during feeding but usually recovered spontaneously. Around day 17 after birth, oxygen saturation repeatedly decreased to approximately 60–80% during lactation and went as low as 31–40%. Therefore, HFNC O_2_ was given again. Even at that, oxygen flow saturation decreased during feeding. Oxygen administration was stopped around day 20 of life. After cessation of HFNC O_2_ administration, the decreases in oxygen saturation during lactation to 50–80% occurred approximately 10 times/day; oxygen saturation usually recovered spontaneously and intermittently with tactile stimulation. The decreased oxygen saturation during lactation persisted until 37 weeks and one day of corrected age, which was 34 days after birth. During sleep, oxygen saturation decreased by 48–85%, 2–3 times/day. Sleep apnea may occur in BOR syndrome, and some cases of sleep apnea are accompanied by abnormalities in the shape of the larynx [6]. The patient’s father stated that, as a child, he sweated a lot when sleeping, but his symptoms appear to have disappeared after surgery in elementary school. At 39 days old, a videofluoroscopic swallowing study (VFSS) was performed after dilution with 20 mL of Omnipaque 300, a water-soluble contrast medium, and 10 mL of normal saline. No evidence of contrast medium collecting in the pharynx or pyriform sinus, no connection to the fistula, and no mass effect was found. No aspiration was found at this time. The decrease in oxygen saturation during sleep was 45–74%, once or twice a day, which recovered spontaneously. The oxygen saturation continued to decrease to approximately 60–70% during breastfeeding more than 4–5 times a day, and 0.3 L/min of oxygen was administered during breastfeeding.

The patient was transferred to a tertiary hospital around day 46 after birth to check for sleep apnoea by polysomnography, rule out abnormalities in the shape of the larynx by laryngoscopy and perform genetic tests for family history. At the tertiary hospital, the brain MRI was normal. Considering the family history, whole-genome sequencing and structural variant calling of the baby, father, mother, and grandmother was performed to diagnose BOR syndrome. Laryngoscopy was not possible due to the size of the child. VFSS and modified barium swallow (MBS) tests showed that the cause of aspiration appeared to be sucking-swallowing-breathing dyscoordination and asymptomatic aspiration. Decreased oxygen saturation during oral feeding was also observed at the tertiary hospital. Thus, the decision was made to continue a diet of 70 mL q 3 h (150 mL/kg/day, 120 kcal/kg/day) using a gavage tube. The patient was discharged two months and seven days after birth (42 weeks and 2 days of corrected age).

## 3. Results

### 3.1. Investigations

Bilateral renal pelvic dilatation was observed during the prenatal examination, but on day three after birth and approximately one month after discharge, the ultrasound showed that the right kidney was 3.1 mm and the left kidney was 4.5 mm, which were within the normal range. Abdominal ultrasonography showed that the cavity was asymmetric at the fundus level of the uterus, and a constricted protrusion was observed in the middle. The cervix and the surrounding space could be seen bilaterally, suspected bicornuate or septate uterus. Figure 4 shows the abdominal ultrasound findings.

Echocardiography performed on days 5 and 1 month after birth showed no specific findings, except for a patent foramen ovale. No abnormal findings were observed during the brain ultrasound on days 3 and 13 and 1 month after birth. The brain MRI was normal. Whole-genome sequencing and structural variant calling of the baby, father, mother and grandmother was performed.

### 3.2. Differential Diagnosis

Laryngoscopy to rule out abnormalities in the shape of the larynx was not possible due to the size of the child.

The VFSS and MBS tests indicated that the cause of aspiration appeared to be sucking-swallowing-breathing dyscoordination and asymptomatic aspiration. During the VFSS and MBS follow-up tests, penetration into the vocal cords was observed before and during swallowing when bottle feeding. The oral transit time was delayed in the oral stage, and a delayed swallowing reflex was seen in the pharyngeal stage. The penetration-aspiration scale score was 5.

### 3.3. Treatment

At five months and 20 days of corrected age (7 months and 13 days old), the second branchial cyst opening and preauricular pits on both sides of the neck were surgically removed at a tertiary hospital. As the left branchial cyst tract reached the tonsil, suture and ligation were performed under the muscle layer.

### 3.4. Outcome and Follow-Up

After discharge from the hospital, the infant was bottle-fed by mouth at home. The caregiver checked the oxygen saturation monitor while feeding the infant and controlled the feeding rate. Oral feeding gradually improved, and the g-tube was removed at an outpatient clinic two months and 14 days after birth. At two months and 27 days old (around 24 days of corrected age), oxygen saturation decreased while feeding formula through a bottle but recovered quickly by controlling the sucking rate. The patient was admitted to the emergency room after vomiting twice the day before because the oxygen saturation dropped to 30% and recovery was not good. No clear aspiration was observed on the chest radiograph. After inserting a g-tube, feeding was continued via gavage, and the patient was discharged. Gavage tube and oral feeding were continued, and at two months and 17 days of corrected age, gavage tube feeding was infrequent. Even during oral feeding, oxygen saturation only decreased to above 70%, and often no decrease in oxygen saturation occurred. At three months and one day of corrected age, almost no decrease in oxygen saturation occurred during oral feeding. Occasionally, if the lactation rhythm was broken at the beginning of feeding, oxygen saturation decreased to approximately 79% but recovered quickly. The patient controlled swallowing and breathing. At six months and 11 days of corrected age, oxygen saturation was no longer monitored during lactation, and no symptoms of gasping for breath were observed, but aspiration sometimes occurred.

The VFSS and MBS follow-up tests were performed at the Department of Rehabilitation Medicine at the tertiary hospital in the upright position. When three milliliters of oral porridge or three milliliters of ground fruit was given with a spoon, no penetration or aspiration while swallowing small amounts was observed. No penetration or suction occurred, even when drinking 3 milliliters of water. However, vocal cord penetration was observed before and during swallowing when bottle feeding. The oral transit time was delayed in the oral stage, and delayed swallowing reflex was observed in the pharyngeal stage. The penetration-aspiration scale score was 5. Therefore, the decision was made to continue the oral diet with caution. The patient was fed in the upright upper body position as much as possible and fed slowly with a spoon while inducing chin tuck. Feeding was stopped immediately if swallowing difficulties were observed and followed up afterward.

During the prenatal examination, bilateral renal pelvic dilatation was observed. However, on day 3 of life and one month after birth, the right kidney was 3.1 mm, and the left was 4.5 mm, which was within the normal range. No specific findings were observed, other than a trend towards decreasing left pelvis AP diameter to 4 mm between 2 months and 19 days and three months and 12 days old. No specific indication of the suspected bicornuate or septate uterus was observed in the ultrasound. The decision was made to follow up on this suspicion. 

The Republic of Korea Bayley Scales of Infant and Toddler Development-Third Edition (K-Bayley-III) evaluation, which was performed at the Department of Rehabilitation Medicine at our hospital at six months and 18 days of corrected age, had a developmental index of 90 points for cognition, 74 points for language and 91 points for motor. The percentiles were: cognitive, 25.2 percentile; language, 4.2 percentile, and motor, 27.4 percentile. The AIMS test performed at eight months and 16 days of corrected age was 50–75 percentile, which was within the normal range.

Whole-genome sequencing and structural variant calling of the baby, father, mother, and grandmother performed at the tertiary hospital showed that the patient, father and grandmother had BOR syndrome. The identified mutation was an *EYA1* exon 5–6 deletion. The results were confirmed with gap-PCR and direct sequencing, showing that 5547 base pairs spanning introns 4–6 and including exons 5 and 6 of the *EYA1* gene were deleted, and one base pair was inserted. Figure 5 shows the genetic test results.

No deletions at this site were found in the population database (DGV, gnomAD SVs v2.1). Deletion of exons 5 and 6 was predicted to result in the loss of 72 amino acids out of 592 amino acids. This mutation was confirmed in the proband, father, and grandmother. Considering the child’s phenotype and family history, this mutation was interpreted as a likely pathogenic variant. No such mutation was found in the patient’s mother.

## 4. Discussion

A diagnosis of BOR can be suspected based on characteristic symptoms. The four major diagnostic symptom criteria are preauricular pits, branchial anomalies (fistulas or cysts), deafness, and renal anomalies. The criteria suggested by Chang et al. differ depending on the presence or absence of a family history [5]. If there is a family history, BOR syndrome can be diagnosed if one or more of the major diagnostic criteria are observed [5]. When there is no family history, BOR syndrome is diagnosed if at least three major findings are present or if two major findings and at least two minor findings, such as abnormalities of the outer, middle, and inner ear; preauricular tags and other abnormalities (facial asymmetry, palatal malformation), are present [5].

Because the phenotype of branchiootorenal spectrum disorder is broad, individuals with the distinctive major findings are likely to be diagnosed using gene-targeted testing, whereas those with atypical features in whom the diagnosis of branchiootorenal spectrum disorder has not been considered are more likely to be diagnosed using genomic testing. [7]. We did not have a targeted gene panel analysis or whole exome sequencing test to diagnose BOR syndrome. At this time, there was a research program, and the patients and their parents were eligible for the study and could take the whole-genome sequencing test.

BOR syndrome is inherited in an autosomal dominant manner and exhibits various phenotypes. Mutations in the Drosophila eyes absent gene (eya), called *EYA1*, which is located on chromosome 8q13.3, are responsible for approximately 50% of cases [8]. The *EYA1* gene plays an essential role in otic placode development between 4 and 10 weeks of gestation [8]. Mutations in the *SIX1* gene can also cause BOR syndrome, but these are much less common. In approximately 20% of cases, a chromosomal rearrangement of 8q13.3 occurs [8]. Submicroscopic deletions, including *EYA1*, can appear as an expanded phenotype, including various musculoskeletal disorders, language development, and developmental delays. This extended phenotype is most often due to deletions of adjacent genes [8].

Common phenotypes of BOR syndrome include hearing loss (90%), preauricular pits (80%), renal dysplasia (65%), branchial fistulas or cysts (50%), anomalous pinna (35%), stenosis of the external auditory meatus (30%), stenosis or aplasia of fistulas (10%) and middle or inner ear malformations [8]. Hearing loss can be sensorineural (25%), conductive (25%) or complex (50%), and symptoms can range from mild to severe [8]. Hearing loss appears from early childhood to young adulthood and sometimes appears suddenly. In some families, hearing loss develops progressively [8]. Malformations of the middle and inner ear (vestibular tract and cochlea) may be present, including displaced, malformed, or fused ossicles and Mondini malformations of the cochlea. Anomalies of the outer ear range from severe microtia to minor anomalies of the pinna (cup-shaped, flattened, or hypoplasia) [8]. The ear canal can be narrow and tilted upwards or malformed, making otoscopy difficult [8]. Renal anomalies range from mild dysplasia (sharply tapered superior poles, blunting of calyces, and duplication of the collecting system) to bilateral renal agenesis with renal failure in approximately 6% of patients [8]. In addition, microphthalmos, narrow and long face, preauricular tag, congenital cholesteatoma, cleft palate, deep overbite, bifid uvula, facial nerve palsy, facial nerve malformation, gustatory lacrimation (the shedding of tears during eating because of misdirected growth of the facial nerve fibers), mitral valve prolapse, congenital hip dislocation, non-rotation of the bowel, pancreatic duplication cyst, euthyroid goiter, benign intracranial tumor, and temporoparietal linear nevus may occasionally occur [8]. In this case, the preterm infant had recurrent sleep and feeding cyanosis with second branchial anomalies. Via videofluoroscopic swallowing study and a modified barium swallow test, penetration into the vocal cords was observed before and during swallowing when bottle feeding.

Repeated aspiration pneumonia has been reported when swallowing is delayed, causing problems with diet or when cyanosis occurs after eating [9]. During embryological development, the posterior belly of the superior hyoid muscle (digastric), which moves the hyoid bone, originates from the second arch. The anterior belly originates from the first branchial arch. Anterior movement of the hyoid bone is necessary to open the upper esophageal sphincter during swallowing properly. In addition, the vertical movement of the hyoid bone is necessary for closing the epiglottis and larynx [10]. Therefore, if there is a second branchial cleft cyst or fistula and repeated cyanosis appears after eating, as in our case, aspiration is highly likely, and VFSS and MBS examinations are necessary [6,9]. The literature does not describe frequent or occasional aspiration symptoms for BOR syndrome. However, from an embryological and anatomical point of view, a second branchial arch anomaly may cause these symptoms. In this case, aspiration was identified as the cause of repeated cyanosis during sleep and meals and after meals via VFSS and MBS tests. After aspiration was identified, feeding was conducted with caution. As the child grows, the degree of aspiration gradually decreased.

During treatment, children with hearing impairments should receive appropriate rehabilitative treatment. Hearing should be tested at least once a year by an otolaryngologist who is well aware of hereditary hearing loss [9]. Otitis media can be treated promptly, and hearing aids can help [9]. Structural defects in the ear may require surgical correction [11]. Branchial anomalies are susceptible to infection and require surgical intervention [11]. In addition, a nephrologist should manage kidney problems with regular check-ups. A kidney transplant may be necessary if a severe kidney anomaly occurs [9].

This case of BOR syndrome was diagnosed early after an accidental discovery of symptoms and family history in a premature infant with a gestational age of 32 weeks and two days at two weeks old. Bilateral branchial cleft cyst and fistula were the points that could be diagnosed early in preterm infants, with rare reported cases around 20 days of age. The prevalence of bilateral branchial cleft anomalies is as low as 2–3%. In the bilateral cases, there was a tendency to have a family history [12]. As a result, the family history was revealed by examining the related syndromes and family histories. Whole-genome sequencing and structural variant calling was performed on the symptomatic baby, father and grandmother, and an *EYA1* exon 5–6 deletion was found, confirming the diagnosis of BOR syndrome. This case is the first report of a preterm infant early diagnosed with BOR syndrome in which the deletion margin was accurately identified by whole-genome sequencing and structural variant calling in Republic of Korea.

## 5. Conclusions

In patients with BOR syndrome, a second branchial arch anomaly may lead to repeated cyanosis during sleep and meals and after meals. The VFSS and MBS tests are essential because aspiration is likely to cause cyanosis.

Bilateral branchial cleft cyst and fistula can be the points that could be diagnosed early in preterm infants or neonates.

This case is the first report of a preterm infant early diagnosed with BOR syndrome in which the deletion margin was accurately identified by whole-genome sequencing and structural variant calling in Republic of Korea.

## Figures and Tables

**Figure 1 children-10-00076-f001:**
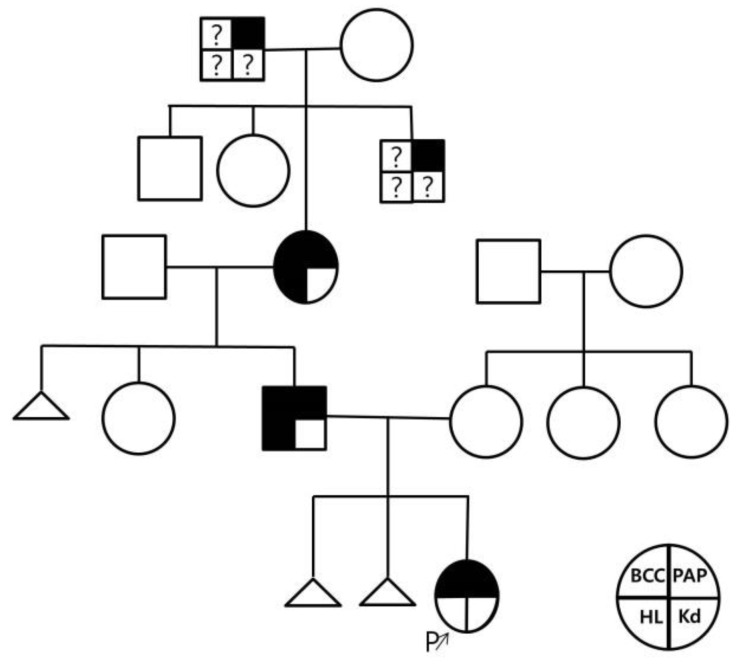
Pedigree of patient with BOR syndrome. BCC: branchial cleft cysts, PAP: preauricular pits, HL: hearing loss, Kd: kidney anomaly, ?: unknown.

**Figure 2 children-10-00076-f002:**
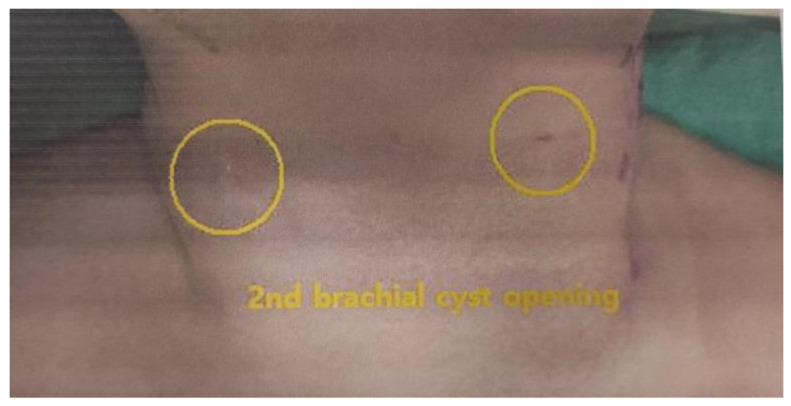
Bilateral second branchial cysts and fistulas opening.

**Figure 3 children-10-00076-f003:**
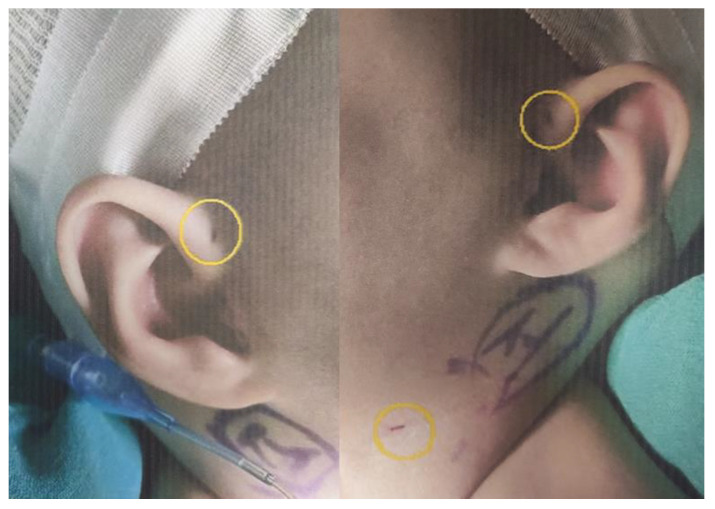
Bilateral preauricular pits and left branchial cyst and fistula opening.

**Figure 4 children-10-00076-f004:**
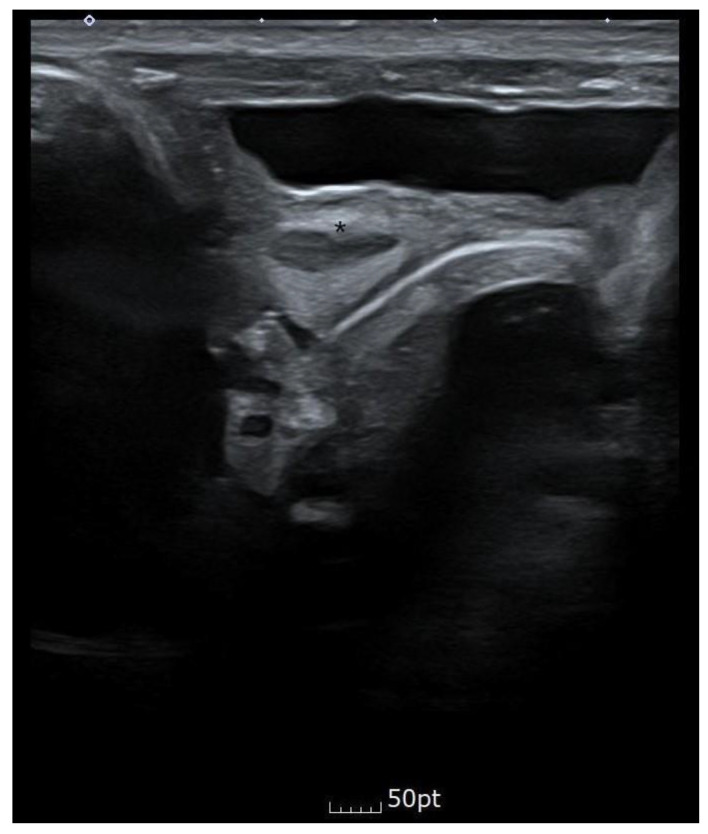
When an abdominal ultrasound transverse section performed with a linear probe was performed, a bicornuate or septate uterus was suspected. Fundus level of the uterus, * a projection in the middle of the uterus.

**Figure 5 children-10-00076-f005:**
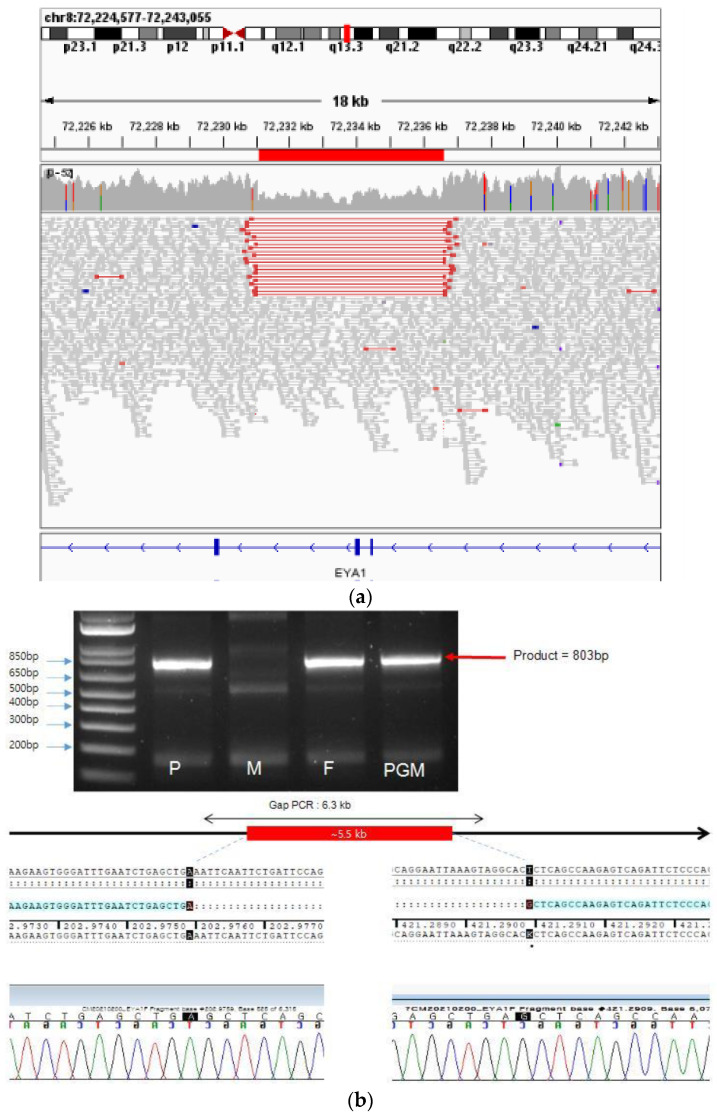
5.5 kb deletion detected in *EYA1* gene. (**a**) Discordant read pair detected in IGV. Suspected deletion spanning exon 5 and exon 6 of *EYA1* is marked in red. (**b**) Gap PCR and Sanger sequencing of the suspected region. The deletion margin is confirmed with Sanger sequencing as NC_000008.10 (NM_000503.6):c.203-2100_419-1133delinsG (chr8:72231057_72236603delinsC). P, proband; M, mother; F, father; PGM, paternal grandmother.

## Data Availability

Not applicable.

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
