# Peer review of "A Preterm Infant with Feeding Aspiration Diagnosed with BOR Syndrome, Confirmed Case by Whole-Genome Sequencing and Structural Variant Calling"

_children, 2022, doi:10.3390/children10010076_

Round 1
Reviewer 1 Report
Thank you for the clinically interesting case report, please confirm the spelling of line 226, which is the explanatory part of Figure 3.
Author Response
the spelling of line 226, which is the explanatory part of Figure 3.
I corrected wrong spelling to " fistula opening"
Reviewer 2 Report
The authors present an interesting case of a preterm infant with classical features of BOR syndrome, including a positive family history. Overall the case report reads long, with a level of detail (e.g. specific ventilatory support settings) that I am not certain contributes much to the overall case. I appreciate the inclusion of a pedigree and a figure showing the sequencing details. I do wonder why the clinicians opted for genomic sequencing instead of first sequencing EYA1 (or a small panel including SIX1 and SIX5), although one could argue that the deletion could have been missed.
A few specific comments:
- Lines 13 & 32 - I imagine this should be "incidence" of 1 in 40,000 newborns
- Lines 35-45 - Much of this detail can be saved for the "case presentation" section (and indeed is included there)
- Line 206 - the gene symbol (EYA1) should be italicized throughout
- Figure 5a obscured the caption for figure 4 in the pdf draft manuscript I reviewed
Author Response
Samsung Seoul Hospital did not have a targeted gene panel analysis or WES test to diagnose BOR syndrome.
At this time, there was a research program as below, and the patients and their parents were eligible for the study and could take the WGS test.
“Collection of clinical and genomic information for the prevention and management of congenital anomalies” funded by the Korea Disease Control and Prevention Agency(2021-ER0706-00).
Also, as you know, the advantage of the WGS test is that it is easier to detect gross deletions than the WES test or targeted gene panel analysis because it has higher detection accuracy of copy number variations.
========================================
- It has been abbreviated as below.(e.g. specific ventilatory support settings)
**Neopuff was initiated at the following settings: peak inspiratory pressure, 20; positive end-expiratory pressure (PEEP), 5 and fraction of inspired oxygen (FiO2), 30%.
=>Positive pressure ventilation was started with the NeoPuff.
**The heart rate recovered to 120 beats 1 min and 40 s after birth.
Flow by O2 was given after positive pressure ventilation was terminated.
=>The heart rate recovered.
**FiO2 of 21% of high flow nasal cannula (HFNC) 4 L/min was applied
=> high flow nasal cannula (HFNC) O2 was applied
**21% FiO2 of HFNC 2 L/min was applied from day 18 to day 21 after birth and then discontinued
=>HFNC O2 was applied from day 18 to day 21 after birth and then discontinued
**Therefore, HFNC flow at 2 L/min FiO2 0.21 was given again
=> Therefore, HFNC O2 was given again
=====================================
- The prevalence is roughly 1 in 40,000.
It's from the book "Smith's Recognizable Patterns of Human Malformation"
- Lines 35-45 - As mentioned, the following content is duplicated and omitted.
- the all gene symbol (EYA1) AND SIX1 have been changed to italics.
- In the PDF manuscript, Figure 5a has been spaced so that
it is not obscured by the caption of Figure 4.